# Working Mode Recognition of Non-Specific Radar Based on ResNet-SVM Learning Framework

**DOI:** 10.3390/s23063123

**Published:** 2023-03-14

**Authors:** Jifei Pan, Jingwei Xiong, Yihong Zhuo

**Affiliations:** College of Electronic Countermeasure, National University of Defense Technology, Hefei 230071, China; panjifei17@nudt.edu.cn (J.P.); zhuoyihong22@nudt.edu.cn (Y.Z.)

**Keywords:** non-specific radar, mode recognition, residual neural networks, support vector machines, multi-source feature extraction

## Abstract

Mode recognition is a basic task to interpret the behavior of multi-functional radar. The existing methods need to train complex and huge neural networks to improve the recognition ability, and it is difficult to deal with the mismatch between the training set and the test set. In this paper, a learning framework based on residual neural network (ResNet) and support vector machine (SVM) is designed, to solve the problem of mode recognition for non-specific radar, called multi-source joint recognition framework (MSJR). The key idea of the framework is to embed the prior knowledge of radar mode into the machine learning model, and combine the manual intervention and automatic extraction of features. The model can purposefully learn the feature representation of the signal on the working mode, which weakens the impact brought by the mismatch between training and test data. In order to solve the problem of difficult recognition under signal defect conditions, a two-stage cascade training method is designed, to give full play to the data representation ability of ResNet and the high-dimensional feature classification ability of SVM. Experiments show that the average recognition rate of the proposed model, with embedded radar knowledge, is improved by 33.7% compared with the purely data-driven model. Compared with other similar state-of-the-art reported models, such as AlexNet, VGGNet, LeNet, ResNet, and ConvNet, the recognition rate is increased by 12%. Under the condition of 0–35% leaky pulses in the independent test set, MSJR still has a recognition rate of more than 90%, which also proves its effectiveness and robustness in the recognition of unknown signals with similar semantic characteristics.

## 1. Introduction

Radar mode recognition is the process of identifying radar functions and working modes, based on parameters and styles extracted from the unknown received signals. With the continuous development of phased array, multi-beam, and signal processing technology, modern multi-function radar (MFR) acquires new abilities. MFR can flexibly change direction and signal style, accomplish stable tracking and monitoring, and can effectively carry out diversified combat tasks such as search, tracking, monitoring, and guidance [1]. Radar mode recognition technology plays an indispensable role in fields like ELINT, cognitive radio, and combat decision-making [2,3]. However, with the radar working style becoming more diversified, the parameter variability is significantly enhanced, and the electromagnetic space becomes more complex. People cannot effectively identify MFR behavior patterns through simple methods such as template matching or statistical histograms [4]. Mode recognition in the modern complex environment has become an important and challenging problem. This paper attempts to propose an effective solution, which combines an artificial intelligence algorithm with radar signal processing knowledge.

Radar behavior mode usually requires corresponding signal style and parameter range, because of its unique functional requirements. The conventional radar receivers receive intra-pulse waveforms, obtained by sampling, as well as the pulse description words (PDW), including radio frequency (RF), pulse width (PW), pulse amplitude (PA), direction of arrival (DOA), and time of arrival (TOA) [5], as shown in Figure 1. Because of relatively pure electromagnetic environment, simple signal parameters, and dull working mode, radar signals could realize behavior pattern recognition by comparison of PDW in previous times [6]. However, with the emergence of complex modulation patterns, such as jitter and stagger, simple parameter comparison has been unable to complete this recognition task. Methods based on statistical analysis and behavioral reasoning have been applied to pulse sorting and recognition [7]. With the development of phased array, radar waveform and styles are more changeable, and the aliasing between parameters is increasingly serious. Therefore, people have turned their attention to the radar intra-pulse waveform, trying to distinguish radar operating modes through waveform differences between signals [8]. However, the wide parameter range and random characteristics of radar signals, make the conventional methods decreasingly effective, when translated from theory to application. The above methods are all based on the analysis and identification of the differences between intra-pulse or inter-pulse data, ignoring the processing laws of the radar itself, that the purpose of radar signal diversification is to better serve the indicators and functions of radar, rather than simple differentiation.

For this reason, this paper proposes a multi-information source recognition and learning framework of radar patterns, called MSJR, which can not only analyze the difference of the extracted signal data, but also calculate the values of functional indicators under typical circumstances, such as detection range, resolution, and ambiguity. It can realize the pattern recognition of non-specific radar, by extracting features from three dimensions: range of parameters, functional indicators, and regularities of data. To this end, we also need to pay attention to a key problem, that is, how to express the complex data regularities, which can be solved through extensive application of deep neural networks.

The main contributions of this paper are as follows:A machine learning method, with radar knowledge embedded, is proposed. The framework creatively applies the radar principles to the dataset, and extracts the radar functional indicators from it, of which the input vector of classifier consists, along with pulse sequences and modulation types, and its feature vector is degraded by the histogram method. Because of the correlation between signal features, MSJR is more beneficial to extract features include the distribution features of data, making the model more extensible and able to identify unknown signals that meet this feature.We propose an MSJR learning framework, based on deep residual network and a support vector machine, for non-specific radar behavior pattern recognition. As an alternative to a single neural network or classifier, the framework optimizes two subnetworks, through model cascades, aiming at pattern recognition accuracy. This framework can better utilize the powerful data expression ability of deep learning, and make use of the classification ability of SVM. Moreover, it is more conducive to identifying patterns in complex environments with serious parameter overlap. Training the MSJR learning framework, can extract the information contained in the signal more efficiently than a single algorithm.The radar operating mode dataset is proposed, with reference to the actual radar, and the proposed method is evaluated on the four types of radar operating datasets commonly used by MFR, with its range and working mode covering most of the information displayed in the public. It also covers modes like target detection, ranging, tracking, speed detection, along with seven inter-pulse modulation types, e.g., constant, jitter, dwell and switch, stagger, slip, wobbulated, and hybrid modulation. In addition, a certain amount of noise, and 0–50% leaky pulses, are added in some cases, which brings more challenges to classification. The experimental results show that the proposed model can achieve more than 90% recognition accuracy under the condition of 10% measurement error and 35% leaky pulse, which is impossible to achieve with previous methods.

The rest of this paper is arranged as follows. Firstly, we review the relevant work in the field of radar operating mode recognition in Section 2. Section 3 describes the radar pattern recognition task and gives the representation method of radar behavior. In Section 4, the MSJR learning framework, based on ResNet24-1D and SVM, is established in four submodules. Section 5 reports datasets, experimental designs, and experimental results, to evaluate and compare the performance of the proposed framework with other recognition technologies. Section 6 concludes the article.

## 2. Related Work

With the diversification of radar systems and the complexity of electromagnetic space, the methods to classify radar signals using the statistical characteristics of signals, such as the histogram method and pulse repetition interval (PRI) transform method [9], have been unable to adapt to the task of radiation source identification and classification. The flexible parameter style and overlapping working range make the statistical characteristics of signals no longer obvious, therefore, mining intrinsic features of signals for recognition and classification, has become the main research direction. These methods can be summarized into two categories: knowledge-driven and data-driven recognition methods.

Knowledge-driven methods exploit the signals according to the known expert knowledge, and extract the internal differences for classification and recognition. The authors of [10,11] proposed high-order statistics of signals, to replace the original signal, for identification after transformation. In [12,13,14], the authors used short-time Fourier transform (STFT), wavelet transform, Wiger–Ville distribution, and other methods, to conduct time–frequency analysis, then they extracted the signal pulse characteristics for identification, which could distinguish the subtle differences of radar electrical signals. The radar ambiguity function was effectively used in [15], which represented the autocorrelation of the signal, and the analysis of the ambiguity function can also accurately locate different radiation sources. As parameters vary quickly, Visnevski tried to describe radar pulses with a multi-level structure in [16], and put forward the concept of “radar word”. He resolved a fixed sequence of pulses into several levels, e.g., radar words, phrases, sentences, paragraphs, to represent radar behavior, and established the relationship between radar words on the basis of a hidden Markov chain. However, it is difficult to solve the contradiction between recognition accuracy and operation complexity caused by the effective encoding length of words [17]. Starting from the semantic relation of radar context, the authors of [18] calculated the confidence interval of unstable evidence based on Dempster–Shafer (D–S) evidence theory, and the identification results exceeded 90%. The common problem of knowledge-driven algorithms, is that the recognition ability depends on the stability of the selected features. Once the prior information is biased, or the selected features do not work on some patterns, the recognition ability will deteriorate greatly. Therefore, more scholars have turned their interests to data-driven algorithms.

Data-driven methods optimize the known model, by learning a large amount of data, to achieve classification and recognition. The widely-used traditional classifier and deep learning network are the typical examples. In [19,20], a clustering algorithm was used to help SVM extract features, but it could not identify complex signal styles. Since deep neural networks are widely used in radar recognition, the authors of [21,22,23,24] used a denoising auto-encoder (DAE), a convolutional neural network (CNN), a residual neural network (ResNet), and a recurrent neural network (RNN), respectively, to achieve their recognition rates of more than 90%, under the given conditions, which verified the advantages of the deep learning method in the field of signal recognition, but at the same time, it also exposed the weakness of a single network facing a complex environment. For the improvement of multi-objective tasks, the authors of [25,26,27] used the method of adding windows in the time domain, to process data, designed the time processing module and the threshold function of the selection window, and thus realized multi-objective classification. The authors of [28,29] proposed a comprehensive recognition method, based on a traditional classifier and deep learning network. It confirmed the feasibility of identifying unknown signals from known signals, which employed a classifier to train the deep learning network and deduce the center vector of the known data.

As for the convolutional residual network and support vector machine model used in the proposed framework, there are many excellent improvement methods in the field of engineering. In view of the difference in results caused by different structures of CNN network, a modified D–S algorithm was designed in [30], which optimizes the network architecture, and achieved higher accuracy in the identification of surface cracks of concrete structures. The work in [31], proposed a comprehensive recognition method that combines long short-term memory networks (LSTM) and CNN, in which CNN was used to extract hidden features in low SNR environments, and LSTM was used to establish the semantic relationship. However, the network could not preserve the global characteristics of the sample, causing the loss of information. In [32], the network adaptively adjusted the activation function threshold according to the input data, and filtered the noise to the maximum extent possible while retaining the original features. However, while removing the noise, the model also removed some valid information. As an excellent nonlinear classifier, SVM’s performance is always restricted by hyperparameters. In [33,34], the regularization parameters and kernel functions of SVM were optimized based on the chaotic differential evolution algorithm and the Black Widow algorithm, respectively. Compared with the conventional artificial fish swarm algorithm and particle swarm optimization algorithm, the regularization parameters and kernel functions of SVM were more effective in thermal process recognition and EEG recognition, but the burden of operation speed was heavier. The authors of [35] proposed a parallel hybrid algorithm for SVM parameter optimization, that combined mainstream sequential minimum optimization with stochastic gradient descent, which achieved a great improvement in operation speed, but its accuracy was not as high as the above algorithms.

In the past work, most researchers have used the means of improving the network model and optimizing the hyperparameters, to increase the signal recognition ability. The methods of extracting target features from deep learning networks and classifiers are essentially data-driven, by dividing the differences of data in different working modes. One of the reasons why it is difficult to further improve the recognition efficiency, is that the data is polluted by noise, and the machine’s learning ability cannot extract more and more effective features. In this case, the recognition ability of the model can be improved by manually defining features, through the description of radar rules. For the task of radar mode recognition, we should pay more attention to the functional differences generated by the signals, rather than the differences between the signals. Therefore, we learn from the authors of [28,36], to encode the radar information, which maps the radar performance indicators into the machine learning model. An integrated recognition framework is used to integrate the advantages of data-driven and knowledge-driven methods. The experiment proved that this idea was feasible.

## 3. Problem Formulation

This paper aims to solve the problem of automatic recognition of radar behavior mode, under complex conditions. This section describes the functional characteristics of the radar behavior mode, and provides the definitions of six modulations, then uses a hierarchical method to establish a mathematical model for radar mode recognition.

### 3.1. Radar Behavior Mode

To meet the functional requirements of radar, MFR needs to optimize the signal parameters of different modes from the aspects of beam scheduling and waveform design. Radar behavior can be limited to four categories: velocity search, ranging, tracking, and search while track [37]. In this study, the airborne phased array radar is taken as an example, to define different modes from the perspective of non-cooperative signals.

#### 3.1.1. Velocity Search (VS)

VS mode is a radar operating mode that can provide the farthest detection range. It mainly detects targets by detecting their speed. This mode uses high PRF at a fixed period, which can maximize the radar coherent processing period, to improve the detection range, and avoid Doppler ambiguity. At this time, the head-on target is in the low clutter area, so sidelobe noise generated by echo ambiguity can be avoided. The target can be detected in the Doppler domain by using constant PRF.

#### 3.1.2. Range while Search (RWS)

RWS mode is the most commonly used search mode of radar, which is characterized by fast detection of multiple targets and provision of range, azimuth, altitude, and other information. According to the repetition frequency, it can be divided into high pulse repetition frequency (HPRF) and medium pulse repetition frequency (MPRF) waveforms. The two waveforms cooperate with each other, to provide rapid detection of a large airspace situation. In HPRF mode, the pulse parameters are relatively fixed, and the range ambiguity is mainly solved by three-stage linear frequency modulation technology. Although the range ambiguity and range illusion can be solved to a certain extent, the accuracy is low. For the MPRF waveform, radar uses dwell conversion or stagger to solve range ambiguity, which improves the accuracy, but decreases the gain of coherent processing and detection range.

#### 3.1.3. Single Target Tracking (STT)

STT mode, is the working mode with the highest accuracy and the most concentrated energy. It can accurately and continuously obtain the target’s azimuth, distance, speed, and other information. The antenna beam can constantly change with the target, to lock it, which can guide the precision weapon to strike. Since this mode aims to track known targets, within the range of the known range gate and Doppler frequency, it does not consider the ambiguity problem, and has the most flexible waveform design method. In order to ensure stable tracking and accuracy for weapons to strike, various modulation, e.g., constant, sliding, jitter, stagger, and periodic, can be adopted in different situations.

#### 3.1.4. Trace and Search (TAS)

TAS is a composite mode, working on tracking, which is mainly used for multi-target tracking in the target airspace, while ensuring a certain searching function. In this mode, the radar utilizes beam scheduling, to ensure stable illumination to the tracking target, ensure that the tracking is not lost, and have high accuracy. TAS has two repetition frequencies: HPRF and MPRF, and has a similar working behavior to RWS when searching, while its tracking mode is similar to STT. The resource division is determined by target number and scheduling method.

### 3.2. Pulse Modulation Mode

A radar signal can be modeled as a collection of discrete signals, with PRI as period. Let T∈P, $n\in Z^{+}$, respectively, be the pulse period and pulse sequence number, then the radar signal, y(t), can be expressed as:(1)$$y\left(t\right)=\sum_{n=1}^{Z}\left(\mu \left(t-nT\right)-\mu \left(t-nT-\tau \right)\right)$$
where μ(t) represents the single pulse signal waveform, and τ represents the pulse width. Using the same method, we introduce the following six modulations and give the mathematical expression of the pulse parameter Y(n), of the *n*th pulse; *M* is the number of pulses in a modulation period. In an ideal environment, the pattern of modulation is clear, but the distribution of data between pulses is very vulnerable to the interference of lost pulses.

The receiver’s reception of electronic signals in a noisy environment is not complete, and the signal intercepted by the receiver may be highly distorted. The signals obtained by electronic reconnaissance are usually mixed signals, of various radiation sources and environmental noise in the receiver frequency band, and radar pulses will be offset and lost to some extent. Due to the limitations of the receiver’s bandwidth and sensitivity, under the limitation of the fixed false alarm rate in the signal detection link, with the decrease in the signal-to-noise ratio, the radar will generate more lost pulses [38]. The degree of lost pulse is affected by receiver performance and the false alarm rate setting. This paper mainly studies the changes in radar pulse descriptors, so it is more appropriate to use a more uniform standard of lost pulse, instead of signal to noise ratio, to measure noise. Figure 2 describes the pulses of six modulations under an ideal environment and with 30% leaky pulses.

#### 3.2.1. Constant

Radar parameters are not completely fixed, and it is generally believed that the variation is less than 1% of the average value, which can be considered as constant modulation, $Y\left(n\right)=\left(1+rand\left(-0.01,0.01\right)\right)\times \sum_{i=0}^{n}Y\left(i\right)/n$. This type of signal has strong stability and is usually used for searching and tracking radars, especially those using moving target indication (MTI) technology and pulse doppler (PD) technology.

#### 3.2.2. Stagger

This mode regularly converts its value around a stable period. $Y\left(n\right)=S_{\left(n\%M\right)}$, S is a set of stagger group sequences. Stagger modulation can eliminate the blind speed of moving targets, and PD radar can also solve range and speed ambiguity, through PRI stagger.

#### 3.2.3. Jitter

The parameter value randomly jitters around a central value. The jitter value is a random sequence, symmetrically distributed in the interval $\left[-T,T\right]$. The relative jitter amount is usually 1%∼30%, $Y\left(n\right)=\left(1+rand\left(-0.3,0.3\right)\right)\times \bar{Y}\left(n\right)$. This jitter feature is usually regarded as an electronic countermeasure against some interference.

#### 3.2.4. Dwell and Switch

The radar pulse sequence is composed of several constant waveforms. Each value stays for a certain time and then converts to the next value. $Y\left(n\right)=D_{\left(n\%M\right)}$, D is a set of variable sequences. High and medium PRF radars usually adopt D&S modulation, to solve the problem of range and speed ambiguity.

#### 3.2.5. Sliding

The value monotonously increases or decreases, and then quickly returns to another extreme value when reaching one, $Y\left(n\right)=Y_{min}+\omega \times \left(n\%M\right)$, ω denotes the sliding change rate. This method can be used to solve the blind distance, and its parameter changes continuously. Sliding modulation can also be used to optimize scanning in angle of pitch, usually the maximum value is less than six times the minimum value.

#### 3.2.6. Wobbulated

The parameter value usually changes periodically, in the form of sine, cosine, or a triangle wave, and its change rate, α, is close to the sine change rate, which can reach 5% of the average value. The change frequency, ω, can reach up to 50 Hz, $Y\left(n\right)=Y_{min}+\alpha \sin\left(n\omega +2\pi \times \left(n\%M\right)/M\right)$. This modulation style is usually used for missile guidance, accurate ranging, and refraining from the shielding effect.

Besides these six widely used modulations described above, there are other modulations such as scheduling, burst, etc. This paper mainly studies the above six modulations and their hybrid forms.

### 3.3. Hierarchical Radar Pattern Recognition Task

According to the hierarchical structure, on which MFR depends, this paper defines the radar signal as four levels, with reference to the “radar word” model, which are, the working mode, processing period, pulse sequence and characteristic parameters. Then, the task of this paper can be expressed as reverse identification of the top-level radar behavior mode, using the set of characteristic parameters received by the bottom layer. Figure 3 shows the hierarchical model of the structure.

**Definition 1.** 
*Working mode B∈R, $R=\left\{A,B,C,D\right\}$, is the set of four behavior modes. B is a set of processing cycle sequences $P_{N}$, $B=\left\{P_{1},P_{2},P_{3}\cdots P_{N}\right\}$, signal processing in the same period is consistent.*


**Definition 2.** 
*Each processing period contains pulse sequences, $p_{n}$, with the same processing method, $P_{N}=\left\{p_{1},p_{2},p_{3}\cdots p_{n}\right\}$, n represents the number of pulses in the processing period, 1≤n≤L, L represents the number of pulses in the maximum processing period.*


**Definition 3.** 
*The pulse is represented by k feature parameter sets, which take PDW and its extended features. Then the ith pulse characteristic set can be expressed as ($RF_{i},PW_{i},PA_{i},DOA_{i},$ $TOA_{i}\cdots parameter_{ki}$).*


The previous research on behavior pattern recognition [36,39,40] mainly considered a single application scenario. The recognition model was established under the condition that the radar model was specific, as well as the fluctuation range of the signal was small, so it had little scalability. In the complex electromagnetic environment, the following three challenges must be solved, to accurately identify the radar operating mode: (1) The signal may come from different radar models, and the unknown signal may be outside the preset range. (2) Because of the change in radar beam and the leaky pulse, caused by low interception technology, the regulation of pulses will be seriously damaged. (3) The difference of radar modulations leads to great deviations in extracting signal features from the defective data. In order to meet the above challenges, and correctly identify the radar working mode, this paper divides the pattern recognition task into three sub-tasks, namely, data regulation extraction, calculation of performance indicators, and feature selection and classification. The regulation extraction will enable the network to focus on the regulations of data distribution, instead of mapping behavior patterns from the heterogeneous overlapping data. The indicator calculation is a human intervention identification process. Based on radar principles, the features expressed in the dataset can be refined to the maximum extent, and interference information can be filtered. The selection and classification of features needs to compare with the real tags, and automatically select the features with the greatest weight, for pattern recognition.

## 4. Proposed MSJR Learning Framework

In order to meet the challenge of pattern recognition of different types of radars, in a complex electromagnetic environment, a new MSJR learning framework embedded radar knowledge, is proposed in this paper. The general diagram of the proposed scheme is shown in Figure 4.

The proposed learning framework mainly includes four steps: (1) ResNet24-1D is used to identify modulation; (2) we extract performance indicators from signal processing, then establish the classification feature sets, to conduct a multi-source fusion model; (3) constant threshold normalization for training and testing; (4) SVM is used to identify behavior patterns.

The reason for the step-by-step implementation, is that the timing information of pulses will be destroyed after feature extraction through modeling, and the step-by-step implementation can make the retained information more complete. It should be noted that ResNet24-1D and SVM are trained in a cascading manner, and ResNet will be jointly optimized, according to the recognition of modulation and the behavior mode.

### 4.1. Residual Neural Network for Modulation Recognition

In the learning framework, we use the deep residual network to solve the problem of initial modulation recognition. A residual neural network, is a neural network that gives consideration to both learning ability and efficiency. It has achieved great success in computer vision and medical image recognition, and is promoted in semantic recognition, signal recognition, and other fields. The application of ResNet for modulation recognition, instead of RNN networks or other new networks, such as Transformer, is mainly based on three considerations:The number of pulses in the sample is 8000∼16,000, so the model needs to have sufficient expression ability to extract signal patterns which are overlapped and seriously missing in long samples;The model structure should avoid being too large and complex, and the cost performance ratio should be considered;It can, not only have a strong representation ability, but also effectively avoid gradient problems caused by deep networks.

ResNet meets the above requirements [41]. The structure of the proposed network model is shown in Figure 5.

The input of the network is the PDW parameter, after leveling, and the sample size is [1 × 8000 × 5]. The network extracts features through a 1D-convolution layer, three residual modules, and a full connection layer, and the filter size of the convolution layer is set to 3 × 1. Each convolution layer is matched with a batch normalization layer, and a ReLU activation function, to solve the problem of gradient explosion and gradient disappearance. After the last residual module, a global average pooling is used, to extract the important features of each channel, and the full connection layer and SoftMax activation function are used to output a [7 × 1] one-hot label. The traditional convolutional neural network, classifies through directly connected convolution layers, this can easily cause model over fitting. The proposed structure uses global average pooling to extract the mean value of each feature map, and directly inputs to the full connection layer. Furthermore, it establishes a quick connection between modules in residual structure, which effectively avoids the occurrence of over-fitting problems, when deepening the network depth. Table 1 details the output shape and parameter number of each layer of the proposed network.

We define the classification loss for modulation classification as: (2)$$L_{c1}=-\frac{1}{N}\sum_{i=1}^{n}\left(y_{i}\ln a_{i}+\left(1-y_{i}\right)\ln\left(1-a_{i}\right)\right)$$
where $y_{i}$ represents the real labels of N modulation style samples in a batch size, and $a_{i}$ represents the predictive labels. $L_{c2}$ is the classification loss for SVM. Trainable parameters in ResNet can be optimized by the combination of $L_{c1}$ and $L_{c2}$, as: (3)$$L=L_{c1}+L_{c2}$$

The following, explains the role of residual connections. Define h(·) as a direct mapping of a residual unit, f(·) as an activation function, then the output of a residual unit is $y_{l}=h\left(x_{l}\right)+F\left(x_{l},W_{l}\right)$, where $x_{l}$ represents the input of *l* layer, $W_{l}$ is the weight matrix of the residual module, $F(x_{l},W_{l})$ represents the value of the residual part, then the output of the next layer is $x_{l+1}=f\left(y_{l}\right)$. For a deeper residual layer, *L*, its relationship with layer *l*, can be expressed as: (4)$$x_{l}=x_{l+1}+\sum_{i=l}^{L-1}F\left(x_{l},W_{i}\right)$$

From the above formula, it can be concluded that any *L* layer can be expressed as the sum of shallower *l* layers and the residual modules between them. Compared with conventional convolutional neural networks, *L* layer is expressed as the product of a shallow characteristic matrix. Applying the chain rule of derivatives, we can obtain the gradient of the loss function L, with respect to $x_{l}$, as follows: (5)$$\frac{\partial L}{\partial x_{l}}=\frac{\partial L}{\partial x_{L}}\frac{\partial x_{L}}{\partial x_{l}}=\frac{\partial L}{\partial x_{L}}\left(1+\frac{\partial }{\partial x_{l}}\sum_{i=l}^{L-1}F\left(x_{l},W_{i}\right)\right)$$

This means that the gradient of *L* layer can be directly transferred to any shallower *l* layer, in the process of back propagation of the network. Because of the additive structure of the residual module, $\partial \sum_{i=l}^{L-1}F\left(x_{l},W_{i}\right)/\partial x_{l}$ cannot be kept at –1 all the time. The model is not prone to the problem of gradient disappearance, so for a better performance in recognition, this will be feasible to train a deeper network.

### 4.2. Joint Modeling Method

The joint modeling module is mainly divided into three feature sets, to form $C=\left\{\vec{\alpha^{w}}⋃\vec{\beta^{q}}⋃\vec{\gamma^{p}}\right\}$, where α is the parameter vector, β is the indicator vector, γ is the modulation rule vector, and the superscripts w,q, and *p* represent the feature dimensions. The main function of the joint modeling module is to extract three feature sets and provide the input data of the SVM classifier.

The first step is to extract the parameter vector. Since the regulation of the signal has been completed in Section 4.1, only the range features will be extracted from the parameter vector. In this paper, the frequency characteristics of PDW in the whole pulse flow, are calculated by the histogram method. δ(n) is defined as a discrete sequence of *N* pulse values. The histogram method is used to obtain the statistical distribution, X(k), of the parameters. The input vector is sampled on X(k), in the interval of 10%, representing the overall information and frequency characteristics of the parameters. In this way, the parameter vector size of each sample is greatly compressed, while preserving the complete information as much as possible. By applying the same process to all parameters, an eigenmatrix, representing the range of parameters, can be obtained.

The next step, is to extract the feature vector, which is a comprehensive expression of radar performance, so it is necessary to use the original parameters and the modulation pattern information identified in Section 4.1. In this paper, radar performance is described by radar detection range, the radar ambiguity function, and ambiguity resolution capability. The performance index only considers the difference brought by the available parameters. For the parameter information that cannot be obtained, typical parameters are used instead. In view of the fact that radars use the reflection of electromagnetic waves to detect targets, the detection range of conventional radars is related to receiver performance, signal modulation, transmission power, and environmental noise [42]. Considering that MFR generally benefits from the gain brought by pulse accumulation, the radar equation can be written as follows: (6)$$R_{m}ax=\sqrt[{4}]{\frac{d_{pw}P_{pk}GA_{e}\sigma nE_{i}}{{\left(4\pi \right)}^{2}S_{min}L_{A}L_{sys}}}$$
where, $d_{pw}$ represents the pulse width, $P_{pk}$ represents the peak power, *G* represents the antenna gain, $A_{e}$ represents the effective aperture of the antenna, σ represents the scattering sectional area, $S_{min}$ represents the reception threshold, $L_{A}$ and $L_{sys}$ represent the propagation and system loss, respectively, and $nE_{i}$ represents the coherent accumulated gain of *n* pulses.

The radar ambiguity function is an important index in designing radar signals. It quantitatively describes the resolution of radar transmitted signals to targets, with different ranges and speeds. Define u(t) as the complex envelope of the transmitted signal. Assume that the time delay is *d* and d+τ, and the Doppler frequency shift is *f* and $f+f_{d}$: (7)$$s_{1}\left(t\right)=u\left(t-d\right)e^{j2\pi \left(f_{0}+f\right)\left(t-d\right)}$$
(8)$$s_{2}\left(t\right)=u\left(t-\left(d+\tau \right)\right)e^{j2\pi \left(f_{0}+f+f_{d}\right)\left(t-\left(d+\tau \right)\right)}$$

Then the mean square deviation of two echoes in the radar matched filter response can be expressed as: (9)$$\varepsilon^{2}=\int_{-\infty }^{+\infty }{\left|u(t-d)\right|}^{2}dt+\int_{-\infty }^{+\infty }{\left|u(t-(d+\tau ))\right|}^{2}dt-2W\left(t\right)$$
(10)$$W\left(t\right)=Re\int_{-\infty }^{+\infty }u^{*}\left(t-d\right)u\left(t-\left(d+\tau \right)\right)e^{j2\pi (\left(t+d\right)f_{d}-\left(f_{0}+f+f_{d}\right)\tau )}dt$$

Order $\bar{t}=t-\left(d+\tau \right)$, then the expression in the integral term in W(t), can be simplified as: (11)$$X\left(\tau ,f_{d}\right)=\int_{-\infty }^{+\infty }u^{*}\left(\bar{t}+\tau \right)u\left(\bar{t}\right)e^{j2\pi f_{d}\bar{t}}d\bar{t}$$

$X(\tau ,f_{d})$ is the ambiguity function of the signal. The larger the value is, the smaller $\varepsilon^{2}$ is, and the harder it is to distinguish the two targets. Figure 6 shows the ambiguity function of the LFM signal.

We use the 4 dB width of X(τ,0) and $X(0,f_{d})$ to represent their nominal resolutions. The range resolution and velocity resolution derived from the nominal resolutions Δτ and $\Delta f_{d}$ of delay τ and Doppler frequency $f_{d}$, are:(12)$$\Delta R=\frac{c\Delta \tau }{2}=\frac{c\int_{-\infty }^{+\infty }{\left|X(\tau ,0)\right|}^{2}d\tau }{2{\left|X(0,0)\right|}^{2}}=\frac{c}{2B}$$
(13)$$\Delta V=\frac{c\Delta f_{d}}{2f_{0}}=\frac{c\int_{-\infty }^{+\infty }{\left|X(0,f_{d})\right|}^{2}df_{d}}{2f_{0}{\left|X(0,0)\right|}^{2}}=\frac{c}{2f_{0}TSP}$$
where *B* is the effective bandwidth of the signal, $f_{0}$ is the carrier frequency, and TSP is the signal pulse accumulation time. Due to the lack of pulse information of the radar behavior pattern in public data, fuzzy function sampling and feature extraction cannot be carried out. Therefore, only theoretical feasibility is analyzed here, and the nominal resolution Δτ, $\Delta f_{d}$ is used as a substitute in the experiment.

In addition to the above indicators, radar ambiguity resolution capability is also an important indicator that we can refer to and obtain. Due to the influence of multiple echoes, and the insufficient sampling rate of low PRF in the Doppler base-band, the radar will have ambiguity in the range domain and speed domain, resulting in uncertainty in target detection. For dwell conversion, stagger, and other modulations, due to its high- and low-staggered PRF distribution, it can effectively solve the ambiguity in range and speed. In general, the joint modeling method, including parameter range, functional indicators, and modulation, can effectively simplify and represent the characteristic information of radar sequence.

### 4.3. Fixed Distribution Normalization

The pulse sample, after joint modeling, contains *M* radar features, and each feature has different units and orders of magnitude, which usually require standardization before feature comparison between modes. The conventional method, is to normalize based on the maximum value of the parameter feature, *K*. For the radar behavior pattern recognition, the test sample sequence may not contain all the behavior patterns. Therefore, normalization based on the maximum value of a certain type of feature may lead to different results for different sample sequences, which will lead to greater deviation in model testing. It is impossible to require an unknown and non-cooperative signal to have a complete pattern sequence. Therefore, this paper normalizes all samples based on a constant distribution. The standardized distribution of the *m*th feature, can be written as follows: (14)$$T_{m}∼N\left(\frac{1}{N}\sum_{i=1}^{N}p_{mi},\frac{1}{N}\sum_{i=1}^{N}\left(p_{mi}-\frac{1}{N}\sum_{i=1}^{N}p_{mi}\right)\right)$$
where *n* represents the number of samples of the feature, and $p_{mi}$ represents the value of the *m*th feature. This method provides a fixed standardized distribution from the training set, and the training and test samples are normalized according to this distribution. Experiments show that this method has the same recognition and classification ability for the samples with leaky patterns. Figure 7 shows the flow of constant threshold standardization.

### 4.4. Support Vector Machines for Mode Recognition

SVM is one of the most widely used kernel learning algorithms, and has achieved good results in the classification of signal, image, biology, and other fields [43]. The support vector machine performs classification by finding the optimal hyperplane in a given sample set. The boundary sample is defined as the support vector. The goal of the classification task is to maximize the distance between the support vector and the hyperplane [44]. Assuming that the sample set $S=\left(x_{i},y_{i}\right),x_{i}\in C,y_{i}\in \left\{\pm 1\right\}$, the optimization problem can be defined as: (15)$$\min_{w}\frac{1}{2}ww^{T}+C\sum_{i=1}^{N}\varepsilon_{i}$$
where *w* is the weight vector of the decision plane, *C* is the penalty variable for error classification, and $\varepsilon_{i}$ is the relaxation variable, indicating the distance that a certain sample is allowed to cross the hyperplane. This is a convex optimization problem. According to the duality principle, the optimal solution can be obtained by using the Lagrange multiplier method. Since the dimensions of $x_{i}$ are $\left|\alpha \right|+\left|\beta \right|+\left|\gamma \right|$, the feature needs to be mapped to a higher-dimensional space, for calculation by using the kernel method. The decision function of the classifier is defined as:(16)$$F\left(x\right)=sigmun\left(\sum_{x_{i}\in s}^{}\rho^{*}y_{i}K\left(x,x_{i}\right)+b^{*}\right)$$
where $\rho^{*}$ represents the Lagrange multiplier of each sample, $K(x,x_{i})$ is the selected kernel function, *s* represents the support vector, and $b^{*}$ is the offset of the model. Different recognition problems require different kernel functions, and linear kernel functions are applicable to situations where the number of features is much larger than the number of samples; The Gaussian (RBF) kernel function is more suitable for the case where the number of samples is sufficient and the data is linearly indivisible. For radar signals, the sample size is much larger than the number of features, and the data meet the linear indivisibility condition, so it is more suitable for the RBF kernel function, the function can be written as:(17)$$K\left(x,x_{i}\right)=exp\left(-\frac{1}{2\sigma^{2}}{∥x-x_{i}∥}^{2}\right)$$

In the selection of SVM hyperparameters, the choice of the RBF kernel parameter, $\sigma^{2}$, and penalty factor, *C*, is important in affecting the generalization ability and training error of the classifier. The larger the $\sigma^{2}$ is, the less sensitive it is to the training data, but the complexity of SVM decreases. If $\sigma^{2}$ becomes smaller, the training error decreases, but the support vector and the complexity increases. In this paper, the improved algorithm of SMO, proposed in reference [35], is used to optimize $\sigma^{2}$ and *C*. The main idea is similar to the coordinate ascending algorithm (CA). This method needs to optimize *N* Lagrange multipliers $\left(\rho_{1},\rho_{2}\ldots \rho_{N}\right)$ in the convex optimization problem, decompose the programming problem with N parameters into multiple quadratic programming problems, and then obtain the optimal $\sigma^{2}$ and *C* through $\rho^{*}$.

## 5. Experiments and Results

This section mainly conducts some simulation experiments and compares the performance of the proposed framework with other algorithms. The test is divided into two parts. In the first part, the convergence and generalization of different neural networks in the learning framework are compared, by taking the PRF modulation as an example. The second part tests the ability of different models to recognize radar behavior patterns in complex environments, and discusses the experimental results. The experiment was run on JetBrains PyCharm 2022 (Prague, Czezch Republic) with Lenovo Intel (R) Core (TM) CPU i7-9900k@4.20 GHz, GPU is NVIDIA GeForceRTX2060.

### 5.1. Dataset

Due to the confidentiality of radar technology, there is no publicly available dataset at present. The radar datasets used in most of the literature only show the range of parameters, which does not closely fit the actual data. Based on [37,39,45,46] and other public materials, this paper constructs two types of datasets, namely, radar pulse sequences set RPDWS-I and airborne multi-function radar behavior mode dataset RPDWS-II, with radar pulse sequences as the main body.

RPDWS-I mainly describes the radar modulation, including constant, jitter, slip, stagger, group change, periodic modulation and mixed modulation. The signal styles are shown in Table 2. RPDWS-II mainly describes the radar behavior modes, including speed search, range search, target tracking and search plus tracking. The modulation mainly focuses on PRI, RF, and PW. The signal style is shown in Table 3. Each signal sample is randomly generated within the given range, not limited to specific models.

### 5.2. Capability of Pulse Modulation Recognition Subnetwork

The experiment was conducted on the RPDWS-I dataset. To verify the effectiveness of the neural network in the learning framework, we trained according to 80% of the training set and 20% of the verification set, and the test set was completely independent of the training set and verification set. The experimental results were compared with the improved CNN network and full connection deep neural network (DNN), proposed in the literature [21], and the overall recognition accuracy was used as the evaluation standard.

#### 5.2.1. In-Training Views

In order to study the influence of different neural network models and network layers on the recognition ability, we have built 6-layer, 12-layer, 18-layer, and 24-layer full connection, CNN, and ResNet networks. After many attempts, the hyperparameters of the optimized network are as follows. The network uses the Adam optimizer, as its structure is detailed in Section 4.1, with its learning rate reduced from $10^{-2}$ to $10^{-4}$, at an interval of 40 epochs, size of the batch set to 128, L2 regularization used in the network, and regularization coefficient set to $10^{-4}$. Each training is 160 rounds in total.

Each neural network is trained for ten times to ensure the stability of training. The training results are shown in Figure 8. The filled part represents the standard deviation of ten times of training. We can see that, after the convergence of the three models, the precision of the CNN and ResNet training sets and verification sets is almost the same, while the precision of the DNN network training set and verification set is quite different. This is because the DNN neurons are directly connected, without the structure of feature extraction, and the average parameter amount reaches an amazing 3G. Although it can fit the training data distribution to a certain extent, the over fitting is serious. From the perspective of model stability, the fluctuation between DNN and ResNet for ten times training is small, while the fluctuation of CNN is relatively serious, which means that CNN needs to be trained repeatedly to find the optimal result.

As shown in Table 4, the average precision of ResNet is 96.3%, much higher than the 79.5% of CNN and 58.3% of DNN. The networks with the highest precision of the three networks is the 12-layer DNN, 18-layer CNN, and 24-layer ResNet. The reason is, that when the number of network layers of DNN and CNN increases, the recognition accuracy has not been improved, but has declined to a certain extent. This is because there is no residual connection in the network, and the shallow training stops, due to the accumulation of small disturbances in deep layers, making the overall recognition rate unable to improve. From the training curve, when there are 40, 80, and 120 epochs, the curve shows obvious fluctuations. This phenomenon is meaningful, because the change in the learning rate on this node makes the model approach the optimal performance again due to the local oscillation. From the above training situation, the performance of ResNet is better than that of DNN and conventional CNN, while blindly increasing the number of network layers may not significantly improve the recognition accuracy, but may lead to performance degradation and the waste of computing resources.

#### 5.2.2. In-Testing Views

In order to further verify the recognition capability of networks in complex environments, we tested three networks, DNN-12, CNN18-1D, and ResNet24-1D, which enjoy relatively good performance in the above analysis. In order to compare the migration performance of the models, we set up two experimental environments, one is to directly train on the original data, and the other is to take certain data enhancement measures, that is 0∼20% of the leaky pulses artificially added by steps. The test set is generated independently of the training set. On the basis of 10% measurement error, 0∼50% missing pulses was set every 5%, to verify the generalization ability of the model.

Similarly, each network selects five models with the best training effect, to ensure the stability of the test. The training results are shown in Figure 9. The filled area represents the standard deviation of the test accuracy. It can be seen that the performance of ResNet24-1D is better than that of the other two networks, under different leakage pulse ratios. Moreover, when the test data and training data have a more similar distribution, the accuracy of the test set is obviously higher. For example, in Figure 9a, when training on the original data with no leaky pulses, the accuracy rate was maintained at a high level. However, the accuracy dropped rapidly after the leaky pulses appeared. In Figure 9b, when training on 0∼20% enhanced datasets, the accuracy rate decreased slowly when the proportion of missed pulses was less than 20%, and ResNet24-1D could maintain a recognition rate of more than 90%.

When the difference between the distribution of the test set and the training set continues to increase, the performance decreases significantly, and some modulation patterns are almost unrecognized. From the results, the method in this paper is lower than the recognition rates of 95% and 96% proposed in [21,24], but it should be emphasized that the data transformation range and error of [21,24] are much smaller than the recognition environment used in this paper, and CNN18-1D, in Figure 9, is established with reference to the parameters given in [21]. It can be seen that the network structure of this paper has more advantages.

Figure 10 compares the confusion matrix of ResNet24-1D in different environments. It can be seen that the recognition performance is relatively stable in an ideal environment and 20% leaky pulses, while a large number of fluctuations occur at 40% leaky pulses. The reason is, that the increase in leaky pulses makes the data distribution of one modulation type tend to other types. In an ideal environment, the six conventional modulation styles are relatively stable, and the wrong classification is concentrated on the mixed style. This is because the mixed style has certain characteristics of other modulation styles, causing confusion in recognition. With 20% leaky pulses, although the absence of a certain pulse affects the distribution of different types of data, the original characteristic shape is basically maintained, so the accuracy is only slightly reduced.

With 40% leaky pulses, nearly half of the parameter values are wrong, causing serious changes to the data regularity. For example, leaky pulses cause the PRI value of adjacent pulses to increase to an integer multiple of the original value. In this way, a large number of constant signals and jittery signals are identified as stagger signals, and more complex signals, such as group changes and periods, tend to be recognized as mixed signals. Surprisingly, stagger, slip, and mixed signals are very little affected by the leaky pulses. This is because, even if the data regularities of stagger and mixed signals are destroyed, the more important thing is to change the signals from one stagger and mixed to another. Even if the PRI value of slip signals changes to an integer multiple within a slip period, the original regularities can still be restored, to a certain extent, during linear fitting of a slip period.

### 5.3. The Performance of the Proposed MSJR Learning Framework

In Section 5.2, we have verified that ResNet24-1D is applicable to the extraction of data distribution rules in the proposed framework. Based on the above work, this section will discuss the performance of the MSJR learning framework in radar behavior pattern recognition, and compare it with the recognition ability of traditional classifiers and deep learning algorithms. The experiment is conducted on the RPDWS-II dataset.

Considering the improvement of recognition performance brought about by the joint modeling method and feature extraction, we identify the leaky pulse signals with different proportions, based on three conditions: direct recognition, multi-source recognition, and recognition using the proposed learning framework. We choose support vector machine with linear kernel, support vector machine with Gaussian kernel, decision tree, random forest, naive Bayes, and other traditional classifiers, to compare with the ResNet deep learning network. Figure 11 shows the recognition accuracy curve of the classifier under various conditions. From Figure 11a–c, the improvement of recognition performance, after taking different measures, is recorded.

In Figure 11a, it is not difficult to see that when the single classifier is used to directly recognize the original radar data, the overall recognition accuracy is low. Even if there is no leaky pulse, the recognition rate is less than 70%. This is because, for different operating modes of the same type of radar, its operating frequency band and pulse arrival direction are relatively fixed, and the information’s amplitude is random. The key parameters for recognition, such as PRI and PW, overlap seriously in the data distribution, as shown in Figure 12. This undoubtedly increases the difficulty of recognition for the classification algorithm focusing on data distribution. However, it can be seen from the comparison of various classifiers, that the ResNet and SVM classifiers have stronger feature extraction ability than other classifiers.

After joint modeling, the overall recognition accuracy of the classifier is improved by about 15%, as shown in Figure 11b. At this time, the recognition ability of traditional classifiers is not much different, but surprisingly, the performance of ResNet is reduced by about 15%, compared with that of direct recognition. The reason is, that after joint modeling, the original data is greatly simplified, and only the feature data extracted from each information source is retained. The sample size is too small, and the difference between the data in the sample is too large, which makes the convolution layer unable to work effectively.

From the performance of each classifier in Figure 11c, when the proposed MSJR learning framework is used, the recognition performance is stable under the condition of 0∼30% missing pulses, and the recognition accuracy has been greatly improved. The recognition accuracy of the linear kernel SVM, Gaussian kernel SVM, decision tree, random forest, ResNet24-1D, and naive Bayes classifiers is 0.788 ± 0.004, 0.941 ± 0.006, 0.822 ± 0.004, 0.835 ± 0.005, 0.944 ± 0.006, and 0.745 ± 0.006, respectively, and the recognition ability of the ResNet24-1D and Gaussian kernel SVM is much higher than that the other classifiers. The reason for ResNet’s performance recovery is, that after the data distribution is standardized, the data in the samples are normalized to the same order of magnitude, which solves the problem of data loss.

Compared with the recognition curves of the three conditions, using the learning framework proposed in this paper, the recognition accuracy of the classifier is higher, and the recognition is more stable when the proportion of lost pulses increases. Surprisingly, under the condition of 10∼50% leaky pulses, the accuracy of ResNet and SVM in classifying radar behavior patterns, is even higher than that of modulation pattern recognition in the first stage. This means that the MSJR learning framework can still accurately identify the working mode in the case of some wrong modulation patterns. This may be that the classifier weakens the influence of modulation pattern recognition errors when extracting multiple information source features for classification, which is beneficial to pattern recognition tasks. It should be noted that the cost performance ratio of the classifier is also an important indicator. In the training above, the parameters of the ResNet24-1D model reached 120 M, and the training time was 2362 s, while the parameters of the SVM model with RBF core were only 4 M, and the training time was 13 s. Therefore, from the perspective of model availability, the SVM classifier avoids repeated construction and training of the large structure deep learning model, and has achieved similar results in feature classification, which may be more suitable for the MSJR learning framework.

In order to compare the recognition ability of the MSJR learning framework to each pattern, the pattern recognition confusion matrix, under the conditions of 20% and 40% missing pulses, is given, as shown in Figure 13. Although the accuracy of pattern recognition decreases with the increase in leaky pulses, it does not mean that all modes are affected. For example, the VS and STT modes are almost unaffected, while the RWS and TAS modes are seriously confused, and their performance degrades significantly, and the RWS mode can hardly be effectively recognized. This is because having 35% leaky pulses makes the range of parameters break the threshold of single waveform to composite waveform. In short, the damage of the waveform will lead to inefficiency in extracting characteristics, and ending up with wrong classifications. This also shows that if the range of radar parameters changes seriously, the performance of the MSJR learning framework is still limited. We will continue to investigate solutions in future research.

Finally, we test five kinds of ILSVRC champion network, in the same environment, to see how the MSJR framework improves performance compared to other similar state-of-the-art reported works. The classical convolutional network models, AlexNet, ConvNet, LeNet, ResNet, and VGGNet have been successfully applied in the field of radiation source identification. The accuracy under different levels of noise, the running time of a single epoch, and the model capacity are compared, and the results are shown in Table 5.

From the analysis of accuracy, AlexNet, ConvNet, and LeNet not only have lower accuracy than the proposed MSJR framework, but also show their shortcomings in the change in accuracy. Due to the lack of effective means to prevent over fitting, the accuracy of the three networks is the highest under the conditions of 10∼30% lost pulses, which is similar to the training set. However, it decreases under the ideal environment, which indicates that the model fails to effectively learn the signal characteristics. The performance of VGGNet decreases seriously under the condition of noise increase, which indicates that its generalization ability is insufficient and its adaptability is weak. ResNet achieved the best performance in the comparison of networks, but its overall performance was about 4% weaker than the proposed MSJR framework, due to the limitations of the purely data-driven approach. From the analysis of processing time and model capacity, AlexNet and LeNet have advantages in this aspect, but their accuracy cannot meet the requirements. ConvNet was inferior in every respect, and its overall performance was insufficient. Compared with the above three models, ResNet, VGGNet and the proposed MSJR framework have deeper network layers, more complex design and larger parameter quantity, so the processing time is slightly higher.

In summary, although the proposed MSJR network is slightly deficient in processing time and model capacity, its comprehensive performance is better than that of the other five networks, and its average accuracy is not only 4∼46% higher than that of other networks, but also has stronger generalization and robustness.

## 6. Conclusions

In this paper, a new learning framework, called MSJR learning framework, is designed for non-specific RMI. As an alternative to a simple neural network or classifier, the advantage is that the overall performance of each sub-module is improved, by the way of cascade training. Meanwhile, MSJR takes into account the impact of radar parameter range, functional indicators, and modulation style, on the behavior pattern, and can more accurately and efficiently extract the behavior features of different information sources within a wide range of overlapping parameters. In addition, the MSJR learning framework uses the ResNet neural network and the RBF core SVM, with the highest cost performance ratio, to realize modulation pattern recognition and pattern classification, respectively. The experimental results show that the model can maintain 90% recognition accuracy, stably, when the leaky pulses are less than 35%, which realizes effective recognition of radar behavior patterns. The stability and generalization of the proposed MSJR learning framework have also been quantitatively verified. Although the recognition performance of the model for RWS mode deteriorates in more severe environments, the MJRS learning framework is generally suitable for identifying non-specific radar behavior modes.

We plan to perform further work on the following three aspects in the future: (1) Working mode recognition under unsupervised or weakly supervised conditions. This makes the model have to find the effective characteristics of the signal by itself. The available methods include, sequence matching, envelope analysis, and prediction state description, etc. No substantial breakthrough has been made so far, but this may be closer to real conditions. (2) Working mode recognition in small samples, or no samples, namely zero-shot learning. The model has the ability to identify unknown signals from the visible training set, which is of great significance for reducing model cost and improving generalization ability. Common methods include, graph convolutional network and encoder-like model. (3) A radar pulse denoising algorithm. This is needed to meet the requirements of sparse space denoising. Available denoising networks include FFDNet, DNCNet, etc. This is helpful for further processing of the signal.

## Figures and Tables

**Figure 1 sensors-23-03123-f001:**
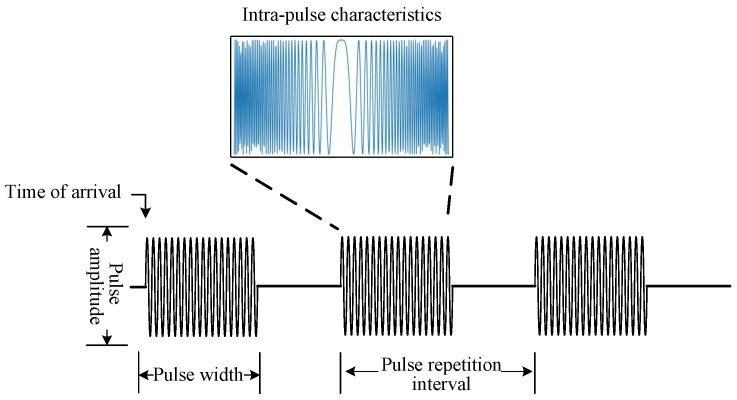
Basic model of radar pulse.

**Figure 2 sensors-23-03123-f002:**
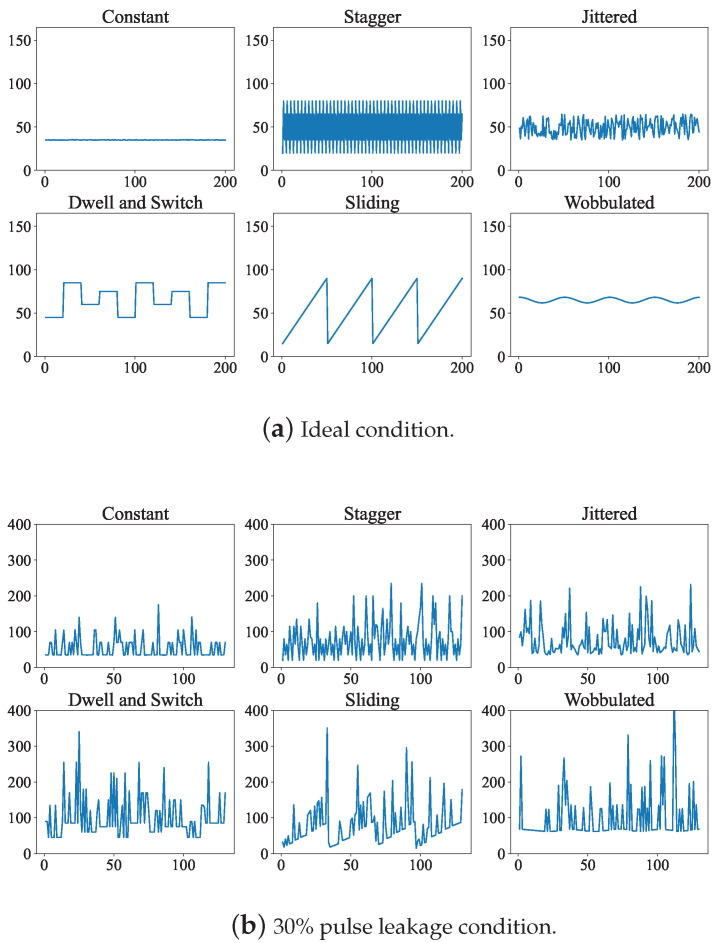
Conventional modulation styles in different environments. (**a**,**b**) respectively show 6 modulation styles under ideal conditions and 30% lost pulse condition.

**Figure 3 sensors-23-03123-f003:**
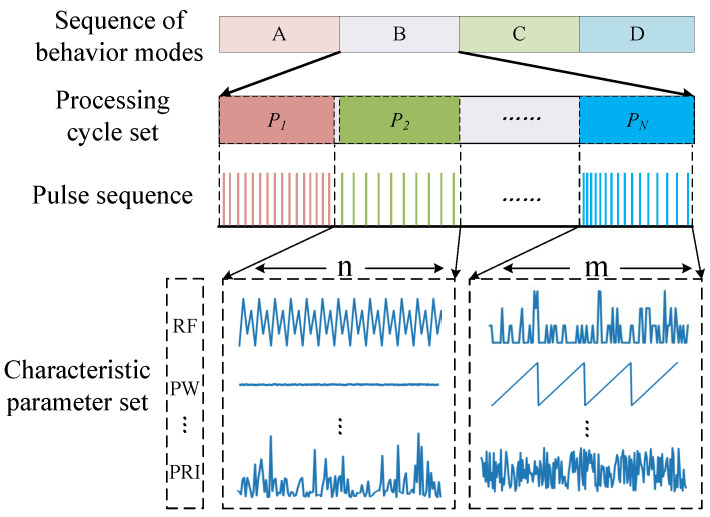
Hierarchical model of radar pattern recognition task.

**Figure 4 sensors-23-03123-f004:**
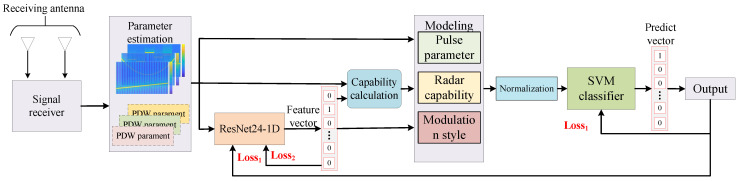
The architecture of the proposed MSJR. In the modeling module, Pulse parameter represents the modeling information of the original pulse parameters, Radar capability represents the calculated performance indicators such as typical detection range, resolution, ambiguity resolution, etc. Modulation style represents the pulse change rules identified by ResNet24-1D, such as constant, sliding, stagger, etc.

**Figure 5 sensors-23-03123-f005:**
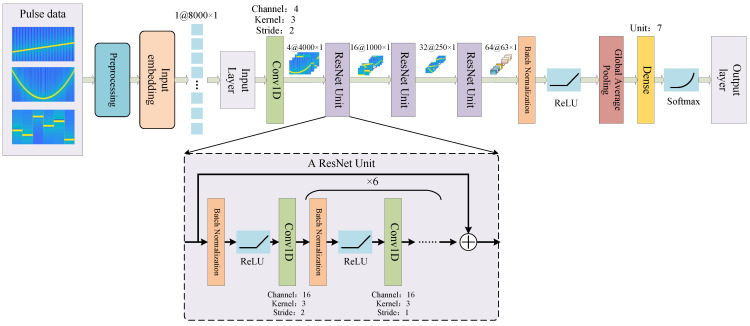
ResNet24-1D deep neural network structure for pulse modulation pattern recognition. The three ResNet Units have the same architecture. Here we only use the convolution of one modulation characteristic graph to simplify the network and save training time. Before the full connection layer, a global average pooling was used to extract the important features of each channel.

**Figure 6 sensors-23-03123-f006:**
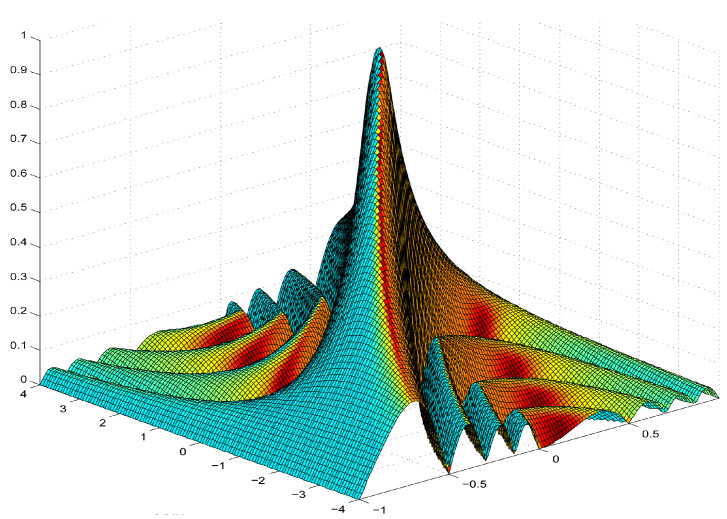
Ambiguity function of linear frequency modulation signal.The color in the figure does not affect the value of the fuzzy function, but only indicates whether it is directly on the Doppler plane.

**Figure 7 sensors-23-03123-f007:**
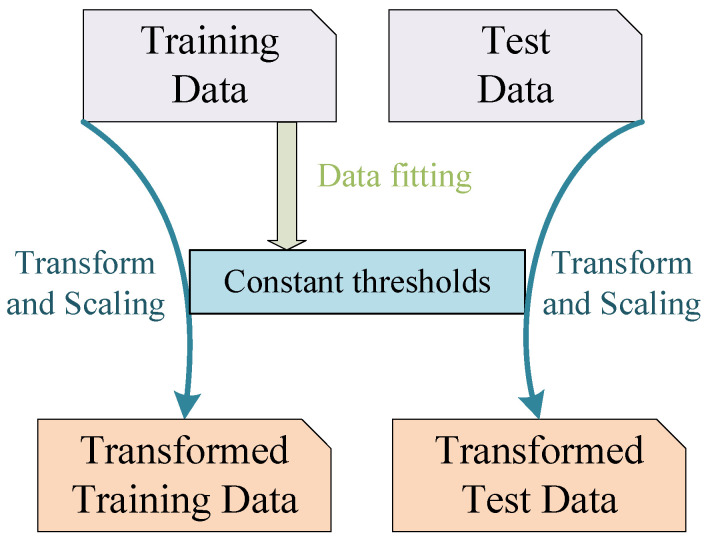
Fixed distribution normalization. The fixed distribution extracted from the training set is used to translate and zoom all datasets.

**Figure 8 sensors-23-03123-f008:**
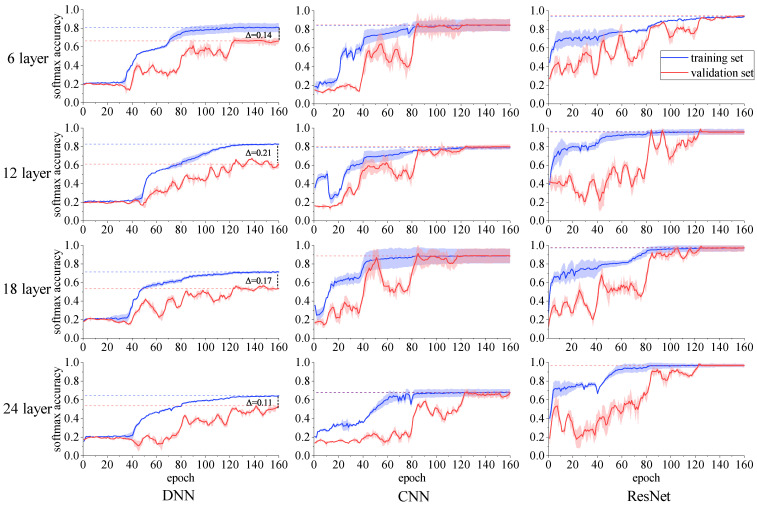
Training accuracy curve of deep learning network. The horizontal line represents the comparison between different networks, the vertical line represents the comparison between networks of different depths, and the filled area represents the standard deviation of ten trainings, to measure the stability of training.The dotted line indicates the reference line.

**Figure 9 sensors-23-03123-f009:**
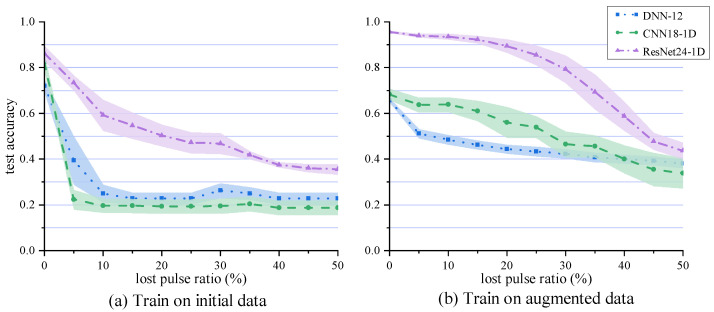
Test accuracy curves of neural networks. Compared with (**a**) the step data enhancement work in (**b**) is adopted during training. The filled area represents the standard deviation of the test accuracy of the same model with the best training effect.

**Figure 10 sensors-23-03123-f010:**
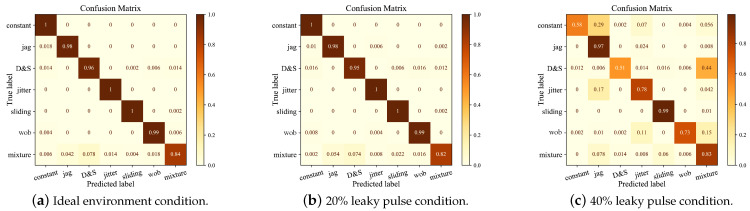
ResNet24-1D modulation mode recognition confusion matrix.

**Figure 11 sensors-23-03123-f011:**
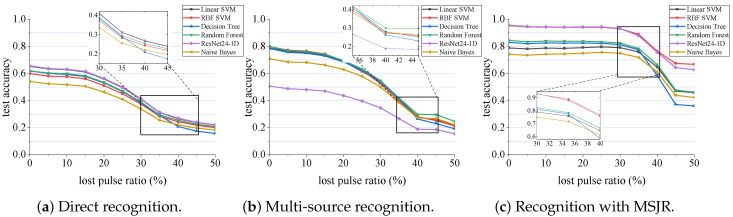
Accuracy curve of mode recognition classifier. (**a**) Directly identified on the original data; (**b**) multiple information sources are used for recognition without feature extraction; (**c**) recognition using the complete MSJR learning framework.

**Figure 12 sensors-23-03123-f012:**
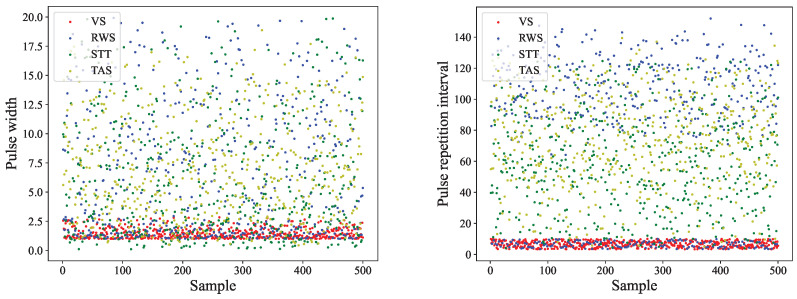
Overlapping of behavior mode parameters.

**Figure 13 sensors-23-03123-f013:**
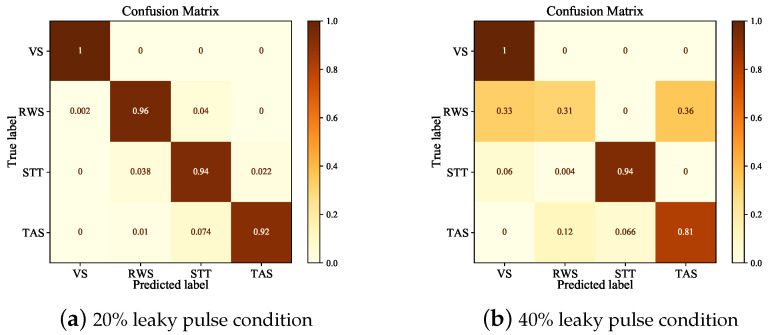
Mode recognition confusion matrix using MSJR.

**Table 1 sensors-23-03123-t001:** Parameters of ResNet24-1D.

Layer	Output Shape	Parameter
Input	8000 × 1	-
Conv1D	4000 × 4	16
ResNet Unit1	2000 × 16	6160
ResNet Unit2	1000 × 32	24,256
ResNet Unit3	500 × 64	94,592
Batch Normalization	500 × 64	256
GAP	64	-
Dense	7	455
Total parameters		125,735
Trainable parameters		123,935

**Table 2 sensors-23-03123-t002:** RPDWS-I dataset. The dataset contains six conventional modulation styles and randomly mixed modulation styles, and each pulse sequence is randomly combined from a given range.

Type	Parameter
Total	Number of pulses = 16,000 Scope of PRI = 3.3∼165 μs PRI perturbation <1% Sample size = 4000
Constant	PRI value = PRImean
Stagger	Number of PRI levels = 2∼8
Jittered	PRI deviation = 10∼30%
Dwell and Switch	Number of pulse groups = 2∼8 Length of pulse groups = 20∼1000
Sliding	PRImax = (2∼6) PRImin Sliding variable cycle = 20∼1000 Direction: up or down
Wobbulated	Wobbulated amplitude = (0.9∼1.1) PRImean Cycle length = 20∼1000
Hybrid	A mixture of any two of the above styles

**Table 3 sensors-23-03123-t003:** RPDWS-II dataset. Each behavior pattern sample is randomly combined from a given range. It should be noted that the test set is generated completely independent of the training set verification set, rather than being divided in the same set of data, to simulate the mismatch between the training set and the test set.

Behavioral Model	Waveform	PRI Value	PW Value	Duty Ratio	Modulation
Total		Number of pulse = 16,000 Scope of RF = 8500∼10,000 MHz PA = PA/PAmax Measurement error < 10% Sample size = 6000			
VS	HPRF	3.3∼10 μs	1∼3 μs	10∼30%	Constant
RWS	HPRF	3.3∼10 μs	1∼3 μs	10∼30%	Constant
	MPRF	50∼165 μs	1∼20 μs	1∼25%	Jittered, Dwell and Switch
STT	HPRF and MPRF	3.3∼125 μs	0.1∼20 μs	0.1∼25%	Constant, Stagger, Jittered, Sliding, Wobbulated
TAS	HPRF and MPRF	3.3∼165 μs	0.1∼20 μs	0.1∼25%	Search waveform refer to RWS Tracking waveform refer to STT

**Table 4 sensors-23-03123-t004:** Average recognition accuracy of neural network.

Number of Layers	DNN	CNN	ResNet
6-layer	65.3%	84.5%	94.7%
12-layer	87.1%	80.1%	95.9%
18-layer	51.9%	88.7%	97.1%
24-layer	49.8%	64.6%	97.4%
Average	58.3%	79.5%	96.3%

**Table 5 sensors-23-03123-t005:** Network performance comparison. The comparison results of accuracy under different noise levels, processing time, and model capacity of AlexNet, ConvNet, LeNet, ResNet, MSJR, and VGGNet. The bold part indicates the optimal value.

Model	Lost Pulse (%)						Process Time (s)	Model Capacity
	0	10	20	30	40	50		
AlexNet	0.407	0.551	0.807	0.561	0.317	0.283	**2.42**	520 K
ConvNet	0.296	0.374	0.7525	0.546	0.397	0.412	11.46	954 K
LeNet	0.416	0.459	0.738	0.481	0.354	0.252	4.16	**285 K**
ResNet	0.890	0.883	0.870	0.847	**0.799**	0.666	10.93	1110 K
MSJR	**0.952**	**0.942**	**0.941**	**0.929**	0.765	**0.770**	14.50	1220 K
VGGNet	0.862	0.479	0.295	0.283	0.272	0.258	9.15	1680 K

## Data Availability

Not applicable.

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
