# Peer review of "Working Mode Recognition of Non-Specific Radar Based on ResNet-SVM Learning Framework"

_sensors, 2023, doi:10.3390/s23063123_

Round 1

Reviewer 1 Report

This manuscript proposed a novel hybridised approach based on machine learning for working mode recognition of non-specific radar, where ResNet and SVM were employed for the task of interest. In the proposed method, ResNet was used to extract features while SVM was used for feature selection. Finally, the performance of the proposed method was validated using experimental data, with satisfactory results. Overall, the topic of this research is interesting, and the manuscript was well organised and written. The detailed comments are given as follows.

1.       The main innovation and contribution of this research should be clearly clarified in abstract and introduction.

2.       Broaden and update literature review on machine learning (such as CNN or SVM) and its engineering applications. E.g. Vision-based concrete crack detection using a hybrid framework considering noise effect

3.       There are several important parameters for both ResNet and SVM, like hyperparameters. Please explain how to optimise them in this research to achieve the optimal recognition performance?

4.       How about the robustness of the proposed method against noise effect?

5.       The running time is also important, and should be considered as a metric for the practical application.

6.       More future research should be included in conclusion part.

Author Response

Thanks for your comments concerning our manuscript entitled “Working mode recognition of non-specific radar based on ResNet-SVM learning framework” (Manuscript ID: sensors-2185379). Those comments are all valuable and very helpful for revising and improving our paper, as well as the important guiding significance to our research. We have studied all comments carefully and have made conscientious corrections. Revised portions are marked in red on the paper. The main corrections in the paper and the responses to the reviewers’ comments are available in PDF below.

Reviewer 2 Report

1. The author should have included a reasonable and convincing comparison table with other similar state-of-the-art reported works. The works described in the table should be critically reviewed and discussed in comprehensive manner in order to show the advantages and innovativeness of the proposed design.. 

2. Most figures require improvement as some of them are too small (i.e.: Fig.2, Fig.3, etc.)

Author Response

(The authors gave the same response as above.)

Reviewer 3 Report

Dear authors,

The contributions of the article must be emphasized in terms of originality, significance, and performance metrics in the abstract.

I strongly recommend the authors to add one paragraph discussing the difference between their work and the previously performed studies in literature.

It is found that some mathematical notations and parameter definitions are missing. Please re-check all the equations and re-define the mission information.

Authors should give more details about implementation, validation procedure, validation time, computational complexity, processing time.

The dynamic properties of the algorithm are not discussed clearly. More details should also be given to this point.

This paper will benefit from a study regarding the effect of parameter variations.

The introduction section provides a comprehensive contextualization of the topic addressed. However, literature contextualization is poor and outdated. For example, about SVM please add:

Least squares support vector machines with tuning based on chaotic differential evolution approach applied to the identification of a thermal process

Recognition of EEG based on Improved Black Widow Algorithm optimized SVM

To have an unbiased view in the paper, there should be some discussions on the limitations of the proposed method.

Is it possible to share the script or data in order to replicate the results?

Best Regards

Author Response

(The authors gave the same response as above.)
